# Proteomic Signatures of Corals from Thermodynamic Reefs

**DOI:** 10.3390/microorganisms8081171

**Published:** 2020-08-01

**Authors:** Anderson B. Mayfield

**Affiliations:** 1Atlantic Oceanographic and Meteorological Laboratory, National Oceanic and Atmospheric Administration, 4301 Rickenbacker Causeway, Miami, FL 33149, USA; abm64@miami.edu or anderson.mayfield@noaa.gov; Tel.: +1-337-501-1976; 2Cooperative Institutes for Marine and Atmospheric Studies, University of Miami, 4300 Rickenbacker Causeway, Miami, FL 33149, USA

**Keywords:** coral reefs, dinoflagellate, global climate change, lipid trafficking, mass spectrometry, predictive modeling, proteomics, symbiosis, upwelling

## Abstract

Unlike most parts of the world, coral reefs of Taiwan’s deep south have generally been spared from climate change-induced degradation. This has been linked to the oceanographically unique nature of Nanwan Bay, where intense upwelling occurs. Specifically, large-amplitude internal waves cause shifts in temperature of 6–9 °C over the course of several hours, and the resident corals not only thrive under such conditions, but they have also been shown to withstand multi-month laboratory incubations at experimentally elevated temperatures. To gain insight into the sub-cellular basis of acclimation to upwelling, proteins isolated from reef corals (*Seriatopora hystrix*) featured in laboratory-based reciprocal transplant studies in which corals from upwelling and non-upwelling control reefs (<20 km away) were exposed to stable or variable temperature regimes were analyzed via label-based proteomics (iTRAQ). Corals exposed to their “native” temperature conditions for seven days (1) demonstrated highest growth rates and (2) were most distinct from one another with respect to their protein signatures. The latter observation was driven by the fact that two Symbiodiniaceae lipid trafficking proteins, sec1a and sec34, were marginally up-regulated in corals exposed to their native temperature conditions. Alongside the marked degree of proteomic “site fidelity” documented, this dataset sheds light on the molecular mechanisms underlying acclimatization to thermodynamically extreme conditions in situ.

## 1. Introduction

Despite the fact that most reef coral-dinoflagellate endosymbioses are environmentally sensitive [1,2,3,4], disintegrating (i.e., bleaching) upon exposure to sub-optimal environmental conditions [5], a number of species/populations in various parts of the world have demonstrated a marked degree of physiological resilience to, for instance, dramatically elevated [6,7,8] and/or highly variable temperatures [9,10,11,12,13,14]. As an example, corals from the southernmost embayment of Taiwan, Nanwan, are subjected to the influence of large-amplitude internal waves (i.e., upwelling [15]) that can cause significant shifts in their abiotic milieu, particularly with respect to temperature. Seawater can fluctuate 9–10 °C on a single day [16], though coral reef ecosystems in this region are thriving (high hard coral cover and diversity [17,18,19]).

Prior studies have attempted to simulate such upwelling-driven temperature fluctuations in the laboratory, using hypothetically stress-sensitive [20] seriatoporid corals (e.g., *Seriatopora hystrix*) from both reefs within Nanwan Bay (“Houbihu”) and “control” reefs <20 km away in the Taiwan Strait (“Houwan”), where upwelling never occurs and where seawater temperatures do not deviate more than 1–2 °C on any given day [21]. It should be noted here that reciprocal transplant efforts in situ have generally failed since the most intense upwelling occurs during typhoon season (boreal summer [22]). Curiously, even corals from non-upwelling reefs were able to acclimate to upwelling-simulating conditions in a laboratory-based reciprocal transplant [21] known as the “*S. hystrix* variable temperature study” (SHVTS), though, in general, both growth and photosynthetic output were higher in corals exposed to their “native” temperature conditions (i.e., upwelling and non-upwelling corals under variable and stable temperatures, respectively).

With these physiological data in hand (summarized in Table 1), a global team of researchers from Taiwan, the United States, the United Kingdom, Canada, Australia, and elsewhere has spent the past decade attempting to gain a better understanding of how these, as well as other Southern Taiwanese coral-dinoflagellate endosymbioses (namely *Pocillopora acuta* and *Stylophora pistillata*) from oceanographically distinct environments, acclimate to changes in temperature in the laboratory [23,24,25,26,27], as well as acclimatize to such environmental heterogeneity and perturbation in the field [28]. As part of these efforts, it was found that variable temperature exposure can thermally harden corals to where they better withstand future increases in temperature [29], a finding later corroborated by other researchers [30].

In analyzing a subset of coral biopsies from these studies, it was found that there is no congruency between mRNA expression and concentration of the respective proteins in Southern Taiwanese *S. hystrix* colonies or their dinoflagellate endosymbiont (family Symbiodiniaceae) populations [31]. For this reason, a series of proteomic analyses were instead undertaken with this coral holobiont to understand how it responds sub-cellularly to changes in temperature [32], as well as how conspecifics from different reef sites (e.g., upwelling [Nanwan Bay] vs. non-upwelling [Taiwan Strait] reefs) vary in their protein biology [27]. A number of proteins were found by 2-dimensional gel electrophoresis (2DGE) followed by mass spectrometry (MS)-based analysis of excised and digested peptides to be down-regulated at variable, upwelling-simulating temperatures relative to stable ones (when pooling data across sites of origin [31,32]). Furthermore, when analyzing samples taken when all corals were at the same acclimation temperature of 26 °C, 14 and 35 host coral+Symbiodiniaceae proteins were maintained at higher levels in corals of Houbihu (>Houwan) and Houwan (>Houbihu), respectively [27] (Table 1).

**Table 1 microorganisms-08-01171-t001:** **A summary of the *Seriatopora hystrix* variable temperature study**. Note that this table reflects a mix of results from published works from the same experiment (see cited references [Ref.].) and this study (“Herein” in the “Ref.” column). When a statistically significant finding (*; *p*<0.05) was documented in the prior reference, the results of the post-hoc test(s) have been shown. Only differentially concentrated proteins (DCPs), and not the “proteins of interest” used in model building (described in the Materials and Methods), have been included in the iTRAQ rows. 2DGE=2-dimensional gel electrophoresis. DEG=differentially expressed genes. GCP=genome copy proportion (a molecular proxy for endosymbiont density). HBH=Houbihu (upwelling site). HWN=Houwan (non-upwelling site). int.=interaction. iTRAQ=isobaric PSII=photosystem II. RNA-Seq=RNA sequencing. RSA=response screening analysis. stab=stable. temp.=temperature. var=variable.

Physiology (Unit)		Temp.	Site	Int.	Post-Hoc Test	Ref.
survival (%)						[21]
growth (mg cm^−2^ day^−1^)				*	higher under “native” conditions	[21]
Symbiodiniaceae density (cells cm^−2^)			*		HWN>HBH	[21]
chlorophyll a concentration (pg cell^−1^)		*		*	higher under native conditions	[21]
maximum quantum yield of PSII (F_V_/F_M_)		*	*		variable>stable, HBH>HWN	[21]
**Biological composition (method)**						
Symbiodiniaceae GCP (qPCR)						[21]
RNA/DNA ratio						[21]
protein/DNA ratio			*		HWN>HBH	[21]
Symbiodiniaceae genotype (qPCR)						[21]
host coral genotype (microsatellites)						[33]
**Gene expression (qPCR)**						
Solaris™ spike exogenous				*		
*rbcL*	endosymbiont	*			variable>stable	[21]
*psI*	endosymbiont	*	*		variable>stable, HBH>HWN	[21]
*pgpase*	endosymbiont	*			variable>stable	[21]
*nrt2*	endosymbiont					[33]
*apx1*	endosymbiont					[21]
*hsp70*	endosymbiont	*			variable>stable	[33]
*hsp70*	host	*			stable>variable	[33]
*actb*	host	*			variable>stable	[33]
*trp1*	host					[33]
*tuba*	host					[33]
*ezrin*	host					[33]
*cplap2*	host			*	HWN-var>HBH-var	[33]
*oatp*	host					[33]
*trcc*	host					[33]
**Transcriptome profiling (RNA-Seq)**					**#DEGs/DCPs**	
	host ^a^	*			1 stable>variable	[31]
	host		*		27 HBH>HWN, 23 HWN>HBH	[31]
	endosymbiont		*		47 HBH>HWN, 9 HWN>HBH	[31]
**Proteins-2DGE**	host	*			97 stable>variable	[31]
	endosymbiont	*			53 stable>variable	[31]
	host		*		9 HBH>HWN, 20 HWN>HBH	[27]
	endosymbiont		*		5 HBH>HWN, 15 HWN>HBH	[27]
**Proteins-iTRAQ**	endosymbiont ^b^			*	1 HWN-stab>all others 1 HBH-var>all others ^c^	Herein

^a^ No endosymbiont proteins were differentially concentrated across temperature regimes by RNA-Seq. ^b^ No host coral proteins were differentially concentrated across temperature regimes by iTRAQ+RSA, yet a number did feature in the proteomic predictive models (see Results.). ^c^ Please note that this was based primarily on the behavior of only two samples.

Despite the fact that nearly 200 differentially concentrated proteins (DCPs) of both host coral and Symbiodiniaceae origin were uncovered in these prior works, it was noted that in many cases, the unique protein spots in the 2-dimensional gels contained a mix of proteins. That being the case, the signal intensity could not be linked to the concentration of any one protein. This makes 2DGE+MS a semi-quantitative method at best, for identifying temperature-sensitive, differentially regulated proteins. Additional data from the MS, such as the peptide “emPAI” scores generated by Matrix Sciences’ Mascot algorithm, can instead be used to infer relative protein concentrations, and label-free proteomic methodologies have indeed improved dramatically in recent years [34]. These methods, though, are arguably better suited at present for work with cell cultures (or model organisms with previously characterized proteomes).

Label-based technologies [35], which instead rely on the conjugation of isobaric tags or stable isotopes to digested peptides, allow for the sequencing (first MS pass) and subsequent quantification (second MS pass) of peptides in a manner that permits the direct linking of a peptide to its concentration in the sample that is akin to next generation nucleic acid sequencing approaches. The more established label-based proteomic approach, known as “isobaric tags for relative and absolute quantification” (iTRAQ; SCIEX, Redwood City, CA, USA) [36], was employed herein to attempt to corroborate prior proteomics-based findings on how the widespread Indo-Pacific reef coral *S. hystrix* responds sub-cellularly to changes in temperature (stable vs. variable/upwelling-simulating). There was also an interest in uncovering proteins whose concentrations differed between corals sampled from upwelling reefs (Houbihu) relative to conspecifics from the nearby, non-upwelling “control” reef (Houwan). More generally, it was hypothesized that this quantitative proteomics approach could uncover thermo-sensitive proteins underlying (or associated with) the molecular basis of coral thermotolerance to highly dynamic temperature regimes.

## 2. Materials and Methods

### 2.1. The SHVTS

The experiment has been described previously [21]. Briefly, corals (*n*=6/site) were collected in May 2010 from both upwelling (Houbihu) and non-upwelling (Houwan) coral reefs of southern Taiwan during a period in which upwelling events were frequently occurring at Houbihu. At both sites, corals were sampled at 26 °C, and the photosynthetically active radiation (PAR) at the depth of coral collection (7–8 m) was ~100 µmol photons/m^2^/s (mean hourly diel PAR [21]). The 12 colonies were fragmented into nubbins (~2 g; *n*=12/colony) and acclimated in indoor aquaria (26 °C; mean hourly diel PAR=90 μmol photons/m^2^/s; maximum PAR=300–350 μmol photons/m^2^/s; sand-filtered seawater) for three weeks prior to exposing half of the nubbins from each site to either a stable, 26 °C temperature treatment (*n*=3 tanks for corals of each site) or one that cycled between 23 and 29 °C over a 6-h cycle (*n*=3 tanks for corals of each site; 12 experimental tanks in total) for one week. For detailed experimental conditions and both laboratory and field seawater quality data, please see [21], and [27], respectively. A number of both physiological and molecular response variables were assessed in the 144 experimental samples (summarized in [27,33], the Introduction, and Table 1) and in general, corals of both sites performed well at variable temperature conditions (even those samples from Houwan never before exposed to such regimes). The genotypes of both Houbihu and Houwan corals were later found to be identical based on analysis of microsatellites [33], and all hosted *Cladocopium* spp. dinoflagellates exclusively [21]. Although future genetic analyses featuring higher resolution approaches, such as next generation nucleic acid sequencing, may ultimately uncover genetic differentiation among the Taiwan Strait (Houwan) and Luzon Strait (Houbihu) corals, clonality has been assumed herein; this signifies that all variation in physiology and proteome biology documented is presumably from environmentally-driven phenotypic plasticity (i.e., acclimation/acclimatization) and not a result of adaptation (assuming microbial assemblages to be similar among them).

### 2.2. Protein Extractions and iTRAQ

After seven days of stable or variable temperature exposure, coral nubbins were sacrificed for a number of response variables (Table 1), and a subset of 12 (2–4 from each site of origin x temperature treatment group) were randomly chosen for iTRAQ. Given the concerns with 2DGE raised above, the proteins previously dissolved in urea rehydration buffer were precipitated in ice-cold acetone, temporarily stored at −80 °C, and then transported in a liquid nitrogen dry shipper at −150 °C from Taiwan’s National Museum of Marine Biology and Aquarium to the MS core facility, where proteins were repeatedly washed with 0.3 M guanidine HCl in 95% ethanol to remove detergents and urea. The washed pellets were dried, resuspended in 0.5 M triethylammonium bicarbonate (TEAB), supplemented with 0.067% sodium dodecyl sulfate (SDS), and prepared for iTRAQ as described in the online supplemental methods (OSM).

Although a sample size of three had been intended for each interaction group, one of the Houbihu-variable (HBH-var) temperature samples was compromised during the washing stage, and so an additional Houbihu-stable (HBH-stab) sample was instead analyzed (Table 2). The 12 proteins (6 from each site of origin and 7 and 5 from the stable and variable temperature treatments, respectively) were dissolved, quantified, run on mini-SDS-PAGE gels (quality control [QC]), concentrated to where all were at the same concentration (15 µg in 30 µL; 500 ng/µL), denatured with additional SDS, reduced, alkylated, digested overnight with trypsin, labeled with iTRAQ reagents (Table 2), quenched, washed thrice with water, dehydrated in a speed-vacuum (speed-vac; Labconco, Kansas City, MO, USA), and resuspended in 20 µL of 2% acetonitrile with 0.1% formic acid (with the latter to aid in ionization). All steps mentioned in this paragraph are detailed in the OSM.

It should be noted that, because there are/were (1) only eight iTRAQ labels and (2) 12 coral protein samples to be analyzed, two iTRAQ batches (hereafter, “A” and “B”) were required. Since some degree of batch-to-batch variation was anticipated, a “normalizer” sample was made by mixing 1–2 µL of each of the 12 target samples into the same tube and concentrating it to 500 ng/µL (the same as the target samples) with the speed-vac. This sample was labeled with iTRAQ reagent 113 (18 µL), run in both nano-liquid chromatography (nano-LC) and MS/MS runs (discussed below), and used as the denominator in calculation of the iTRAQ protein concentration ratios (iTRAQ data are always presented as ratios to a pre-set “control” sample.).

### 2.3. Nano-LC-MS/MS and Data Pre-Processing

Labeled proteins were analyzed by nano-LC and MS/MS as in the OSM. The *S. hystrix*-Symbiodiniaceae holobiont transcriptome [31] (as a fasta file containing all contigs from the host corals, Symbiodiniaceae dinoflagellates, bacteria, viruses, and non-dinoflagellate eukaryotes) was queried with Proteome Discoverer (ver. 2.2; Thermo-Fisher Scientific, Waltham, MA, USA) to assign a protein identity to the MS-derived spectral data generated using a false discovery rate (FDR)-adjusted *q*-value of 0.01. This effort contrasted with prior proteomic analyses of thermally challenged corals, in which either the cellular behavior of only one eukaryotic compartment was profiled in isolation or the reference nucleic acid library used for protein identification was derived from different samples from those analyzed by proteomics (thereby limiting the number of proteins that could be identified with confidence) [37,38]. All nucleic acid-related (i.e., RNA-Seq) bioinformatic analyses associated with this project, as well as the capacity to BLAST and search with MS files against the *S. hystrix*-Symbiodiniaceae transcriptome, can be found on the interactive website: http://symbiont.iis.sinica.edu.tw/s_hystrix/static/html/. Although the website’s “MS-SCAN” feature can be used to assign protein identities to MS spectral data without the need for pre-processing on Proteome Discoverer, it does not yet have the capacity to distill or interpret iTRAQ data [31,32]. For this reason, the transcriptome server was generally relied upon exclusively for extracting mRNA expression data corresponding to the proteins sequenced (see treatise on mRNA vs. protein correlation analysis below.). It should be mentioned here that all proteomic data associated with this manuscript have been included in a supplemental, tab-delineated (Excel) spreadsheet (the online supplemental data file [OSDF]), and the raw data (as RAW, MZML, and MZID files) have been deposited on the University of California San Diego’s (USA) MassIVE data repository (accession: MSV000085863, doi:10.25345/C5P46C), Proteome Xchange (accession: PXD020679), and NOAA’s National Centers for Environmental Information (accession: 0216077).

### 2.4. Data Filtering, QC, and DCP Identification

QC filtering of the identified peptides was undertaken as in the OSM. Briefly, only labeled peptides >6 amino acids (AA) and <140 AA that were found in both iTRAQ batches were considered. A variety of univariate and multivariate statistical analyses were undertaken with the 30 proteins that passed all QC; details can be found in the OSM. In short, JMP^®^ Pro’s (ver. 14; Cary, NC, USA) “response screening” analysis (RSA; FDR-adjusted) was used to search for proteins whose concentrations differed significantly among site of origin (df=1), temperature treatment (df=1), and/or their interaction for the following four groups of proteins: unknown compartment (*n*=3), bacterial+viral (*n*=4), endosymbiont (*n*=12), and host coral (*n*=11). Please note that this signifies that the FDR differed across compartments, though the same FDR-adjusted alpha (0.01) was set for all four. This ensured that the probability of committing a type I error did not very between compartments of lower (bacterial) and higher (host coral) biomass. Putative, RSA-screened DCPs were checked for normality, transformed when necessary, and then re-analyzed by 2-way ANOVA (site vs. temperature) to corroborate RSA findings.

### 2.5. Multivariate Proteomics

Principal components analysis (PCA; on correlations) and multi-dimensional scaling (MDS; on standardized data) were carried out to depict the relationship and similarity among samples, respectively, with the composite, 30-protein dataset, as well as with the host coral (*n*=11 proteins) and endosymbiont (*n*=12) datasets alone. Since it was hypothesized that the proteomes would differ across corals from the two sites of origin, a k-means clustering algorithm from JMP Pro was programmed to sort samples into one of two clusters. When a sample clustered with the wrong site, it has been indicated as such in the corresponding PCA biplot. Finally, permutational MANOVA (PERMANOVA) was used to uncover multivariate effects of site of origin, temperature treatment, and their interaction on the entire proteome. Unlike MANOVA, PERMANOVA can detect multivariate treatment effects when there are more response variables (*n*=30 proteins herein) than samples (*n*=12). PERMANOVA and k-means clustering were conducted with the composite, 30-protein dataset only. It is worth mentioning that, because of there being only two samples from Houbihu exposed to the variable temperature treatment (which resulted in an unbalanced design of *n*=2, 3, 3, and 4 for the four interaction groups), the PERMANOVA had low power to detect a statistically significant interaction effect. Therefore, PERMANOVA was primarily relied upon for documenting main effects (site and temperature), with PCA and MDS used to qualitatively resolve (1) inter-sample differences and (2) putative interaction effects.

### 2.6. Proteomic Data Modeling

In addition to the FDR-adjusted RSA method for unveiling DCPs, two statistical modeling approaches were undertaken to (1) corroborate the RSA-screened DCPs and (2) build models capable of predicting coral behavior (response to temperature treatment for a given site of origin). First, JMP Pro’s stepwise discriminant analysis (SDA) platform was used to identify the protein(s) that featured in the most parsimonious predictive model that correctly classified each of the 12 samples by site of origin, treatment, and their interaction with 100% confidence. The model was verified by training and validating random subsets of samples in several million iterations (see OSM for details.), and since all models featured fewer than 11 proteins, Wilks’ lambda could be calculated to demonstrate whether the multivariate mean of the selected “proteins of interest” (POIs) differed significantly across site, treatment, or their interaction (MANOVA alpha=0.05). Please note that, throughout the manuscript, only those proteins identified by RSA are referred to as DCPs; those proteins whose concentrations did not vary significantly in isolation of other proteins yet featured in statistical models that could predict coral behavior with respect to the experimental treatments are instead referred to as POIs.

As a secondary means of selecting POIs, and because information theory-based approaches are arguably better suited for analysis of ‘OMICs datasets than more traditional, inferential statistics that are rooted in the calculation and interpretation of *p*-values [39], JMP Pro’s stepwise regression (SRA) platform was used with the experimental factor of interest (site, temperature, or their interaction) as the Y and the 30 proteins as predictors (X). The AI model was built in a stepwise, forward fashion such that the Bayesian information criterion (BIC) was minimized. Unlike RSA, SDA and SRA were not carried out individually for each compartment because both PCA and MDS revealed that combinations of proteins from all compartments (not any one in isolation) best partitioned data by experimental factor (see Results.). Although SRA is based on information theory criteria and is not strictly governed by parametric statistical assumptions, the ancillary MANOVAs calculated as part of the SDA should be interpreted in a parametric context. For datasets with small sample sizes, such as this one, the resulting reduced-parameter MANOVA for the site x temperature interaction may well be characterized by a 0% misclassification rate and a highly significant *p*-value despite the hypothetically low associated power. However, such significant SDA-based MANOVA interaction effects must be interpreted cautiously given that only two samples were in the Houbihu-variable temperature group.

## 3. Results

### 3.1. Overview of the Sequenced Proteome

In iTRAQ batches A and B (OSDF), 2014 and 3792 peptides, respectively, were mapped to the conceptually translated SHVTS transcriptome (derived from 14,299 and 24,381 mass spectra, respectively). Of these 5806 peptides, 556 were sequenced in both batches (Appendix A; *n*=5250 uniquely sequenced peptides), and of the 278 (14%) and 409 (11%) labeled proteins in batches A and B, respectively (Appendix A), 30 were found in both. These 30 proteins were the focus of this work given that they passed all QC criteria. In terms of their compartmental breakdown (Figure 1a), 11, 12, 3, 1, and 3 were of host coral (37%), Symbiodiniaceae (40%), bacterial (10%), viral (3%), and unknown origin (10%), respectively. With respect to their functional breakdown (outer-most wedge of Figure 1b), 8 (27%) could not be identified based on alignment-based homology searches against the conceptually translated SHVTS transcriptome, NCBI, UniProt, pFAM, or KEGG databases (via tBLASTn or BLASTp). The remaining 22 proteins spanned a number of cellular processes, including DNA packaging, replication, and editing (*n*=5), Golgi-associated lipid and protein trafficking (*n*=3), cell-cell interactions (*n*=2), protein–protein interactions and protein homeostasis/QC (*n*=2), the stress response (*n*=2), and photosynthesis (*n*=2). For the coral host specifically (middle wedge of Figure 1b), proteins involved in the aforementioned DNA processes and the stress response were most commonly sequenced, whereas proteins involved in photosynthesis, Golgi lipid+protein trafficking, and protein QC were most common cellular processes represented in the 12-protein endosymbiont dataset (inner-most wedge of Figure 1b).

### 3.2. Multivariate Proteomic Analysis

When looking at these 30 proteins in a multivariate context (Figure 2), both PCA (Figure 2a) and MDS (Figure 2b) depicted some separation by site of origin; the exception is the sample Houwan-variable-tank V1 (HWN-var1), which fell within the Houbihu cluster (confirmed by k-means clustering). The site of origin effect was mainly driven by the corals exposed to their native temperature conditions; Houbihu-variable and Houwan-stable samples were best distinguished from each other by PCA and MDS, whereas the transplanted corals were more similar to each other. It should be mentioned that, because data were standardized prior to MDS to give each protein equal weighting (i.e., highly concentrated proteins given the same weight as low-concentration ones), the MDS and PCA results were highly similar.

This observation that the site of origin effect on the proteome was more pronounced than temperature effects was partially supported by PERMANOVA (Appendix A); although neither temperature nor interaction effects were documented, a marginal (*p*=0.06) difference between corals of the two sites was detected. When conducting PCA and MDS with the 11 host coral (Figure 2b,c, respectively) and 12 Symbiodiniaceae proteins (Figure 2d and Figure 2e, respectively) separately, the site of origin differences were more poorly resolved. That being said, the stable temperature treatments of Houbihu and Houwan tended to be well divided by MDS (Figure 2f) for the endosymbiont proteins. In most biplots, stable (S) and variable (V) temperature samples generally appeared intermixed; however, there was some separation between stable and variable temperature samples in the Symbiodiniaceae MDS (Figure 2f), the exception being one Houbihu-stable temperature sample positioned near the bottom of the chart.

### 3.3. DCPS

The AI-based RSA identified four putative DCPs (Table 3 and Table 4, Figure 3): two, one, and one of Symbiodiniaceae (Figure 3a,b), unknown (c64657_g1; Figure 3c), and bacterial (Figure 3d) origin, respectively (Figure 1c). Although the unknown protein (Figure 3c) was maintained at 2.5-fold higher levels in the Houbihu-variable samples, this difference was not corroborated upon log-transforming the data to achieve more normally distributed residuals. The bacterial nucleotidyltransferase (DNA replication) protein was maintained at 40-fold higher levels in samples from Houbihu exposed to variable temperatures (Figure 3d). However, upon finding the distribution to be skewed, a non-parametric 2-way ANOVA did not corroborate this finding. Furthermore, since the microbiomes of the coral samples were not characterized, it could not be known whether this was simply an artifact due to samples from Houbihu having differing bacterial assemblages. It is worth noting here that the relative bacterial protein signal, however, did not differ significantly across the experimental factors (2-way ANOVA of pooled bacterial protein concentrations, *p*>0.15). For thess reasons, although this bacterial protein was useful in model building (discussed below), mechanistic inferences were not drawn from it. The two DCPs of endosymbiont origin, sec34 (Figure 3a) and sec1a (Figure 3b), are both involved in Golgi lipid and protein trafficking (Figure 1e), yet they showed contrasting trends. sec1a (Figure 3b) was maintained at highest levels in Houbihu samples exposed to variable temperatures (2.5-fold higher than the pooled mean of the other three interaction groups); however, please note that, although this protein passed the FDR-based RSA, no pairwise, post-hoc differences were documented with the box-cox-transformed data. In contrast, sec34 was concentrated at three-fold higher levels in Houwan samples exposed to stable temperature (i.e., both proteins higher under native temperature conditions), and this pairwise difference was supported by post-hoc testing using an honestly significant difference threshold matrix and a *q*-value of 3.20 (*p*<0.05).

### 3.4. SDA

Three of these four “most differentially concentrated” proteins (excluding unknown protein c64657_g1) were generally useful in model building, as well (Figure 4, Table 3). The Symbiodiniaceae sec34 protein, along with unknown protein c103260_g1, were incorporated into an SDA model (Figure 4a) that was able to predict the site of origin of each of the 12 samples with 100% confidence (i.e., misclassification rate of 0%). When building an SDA model for temperature treatment, the Symbiodiniaceae sec1a protein was instead more important in predicting coral behavior with respect to stable or variable temperature exposure; the same unknown protein c103260_g1, as well as a host coral calmodulin (c65095_g1), also featured in the temperature SDA model (Figure 4b). For the SDA interaction of site and temperature (Figure 4c), four proteins were required to build a model that could correctly identify the site of origin x temperature treatment interaction group for each of the 12 samples with a 0% misclassification rate; this model included the endosymbiont sec34 protein, as well as three proteins not featured in any other SDA model: a Symbiodiniaceae peridinin chlorophyll a-binding (PCP) protein (c79881_g2), a Symbiodiniaceae “in between ring fingers” (IBRF) protein (c104_g1), and the bacterial nucleotidyltransferase (c83543_g1; an RSA-identified putative DCP Figure 3d). The corresponding MANOVAs were statistically significant for all three models (see Wilks’ lambda-associated *p*-values within Figure 4.).

### 3.5. SRA

SRA (Table 3) yielded similar suites of proteins that were incorporated into simple (*n*=2–4 proteins), parsimonious (low-BIC) models capable of explaining 100% of the variation in the dataset (R^2^=1.0 for all models). The two, aforementioned Symbiodiniaceae Golgi trafficking proteins explained the greatest proportion of the variation in the Houwan vs. Houbihu (site of origin) and stable vs. variable temperature comparisons (Table 3). The unknown protein identified by SDA, c103260_g1, also featured in the temperature and interaction models, and the SDA-identified Symbiodiniaceae IBRF was included in the site of origin model. Three POIs not uncovered by RSA or SDA were found in the SRA model of the site x temperature interaction effect: a different Symbiodiniaceae PCP protein (c79881_g1), a host coral Pao retrotransposon peptidase (c197443_g1) involved in DNA modification, and a host zinc finger CCCH domain-containing protein 3-like (c75958_g1) involved in exporting mRNA from the nucleus. In total, then, there were four DCPs uncovered by RSA and seven POIs uncovered from SDA and SRA. See Appendix A for a Venn diagram depicting the congruency among RSA, SDA, and SRA for selecting differentially regulated proteins that were useful in predicting/modeling coral responses.

### 3.6. Breakdown of DCPs and POIs

In terms of the compartmental (Figure 1d) and functional breakdown (Figure 1f) of the 4 DCPs+7 POIs, 5 of the 11 (45%) were of endosymbiont origin, versus only 3 from the coral host (27%). Although this proportion did not differ significantly from the 12:11 endosymbiont:host protein ratio (Figure 1a), the 5:3 DCP+POI ratio was significantly higher than the biomass ratio of this coral (~1:2 [31]; Fisher’s exact test, *p*<0.001). In other words, the dinoflagellates constitute ~1/3 of the holobiont’s biomass yet contributed 45% of the most variable proteins. Of the 11 DCPs+POIs, the dominant gene ontology cellular processes (Figure 1f) were Golgi lipid and protein trafficking (*n*=2; both of endosymbiont origin), DNA processes (one host coral and one bacterial) and photosynthesis (*n*=2; both of endosymbiont origin). Golgi trafficking proteins were relatively enriched in the DCP pool versus the entire 30-protein dataset (Figure 1e). 

### 3.7. Gene vs. Protein Correlation Analysis

For a subset of six samples (Table 2), both RNA-Seq transcriptome profiling (Illumina) and iTRAQ-based proteomics were undertaken. Separate linear regression analyses were performed for the 30 proteins that passed QC versus their respective mRNA levels (obtained from the interactive SHVTS transcriptome server discussed above), and the mean R^2^ of 0.022±0.242 (standard deviation for this and all other error terms unless noted otherwise) was not significantly higher than 0 (*z*-test, *p*>0.05); please see Figure 5 for a distribution of the correlation coefficient values. When looking on a molecule-by-molecule basis, there were no statistically significant positive correlations for any gene/protein pair. That being said, there were some modest, positive, linear correlations (Figure 5; five molecules were characterized by R coefficients >0.5.); these may have not been significant only because of the low sample size. Mean host coral (*n*=11 genes/proteins), endosymbiont (*n*=12), and bacterial+viral (*n*=4) R coefficients were 0.131±0.398 (R^2^=0.017), 0.120±0.470 (R^2^=0.014), and 0.237±0.454 (R^2^=0.056), respectively, and did not differ significantly from one another (one-way ANOVA effect of compartment, *p*=0.947). Despite the lack of correlation between gene expression and the concentration of the encoded proteins, the marginal site of origin effect documented at the protein-level (Figure 2 and Appendix A) was corroborated at the mRNA level when using a data complexity reduction approach known as t-distributed stochastic neighbor embedding (Appendix A).

## 4. Discussion

In contrast to what had been hypothesized, few proteins differed significantly in concentration between temperature regimes. Instead, the four most differentially concentrated proteins tended to show an interaction effect in which concentrations were maintained at higher levels in samples exposed to native temperature conditions (Table 4): upwelling corals exposed to upwelling-simulating (i.e., variable temperature) conditions and non-upwelling corals exposed to non-upwelling (i.e., stable temperature) conditions. Since one of these proteins could not be assigned a function or compartment of origin and the concentration of another may have been linked to differential microbial assemblages among samples (addressed above), the discussion has been focused on the two remaining proteins, both endosymbiont sec proteins involved in Golgi-mediated lipid vesicle and protein trafficking. sec34 and sec1a were maintained at significantly and marginally higher levels, respectively, in endosymbionts within corals from Houwan and Houbihu, respectively, exposed to stable and variable temperature conditions, respectively; this is also evident from the PCA, in which the two native interaction groups, Houbihu-variable (*n*=2 only) and Houwan-stable, are most distinct from each other (with the sec proteins being characterized by high PC1 eigenvalues). However, please note that, although the overall model interaction effect was significant for sec1a, there were no statistically significant, pairwise differences upon box-cox-transforming the non-normally-distributed data.

In a prior 2DGE work with these same samples, it was found that proteins involved in endosymbiont lipid body (LB) formation, structure, and maintenance (namely caleosins and oleosins) were down-regulated upon a week-long exposure to a variable temperature regime [31]. Although neither caleosin nor oleosin was in the final 30-protein dataset herein, the uncovering of thermo-sensitive LB and lipid trafficking proteins by both 2DGE and iTRAQ may implicate a role of Symbiodiniaceae lipid metabolism, and perhaps even translocation, in the response of the *S. hystrix* holobiont to differing thermal regimes. Of particular interest is the up-regulation of sec1a in corals from Houbihu exposed to variable temperature conditions since this finding was statistically significant by RSA and associated with the wide partitioning of the Houbihu-variable samples from the remaining 10 corals (depicted by PCA and SDA, though see caveats raised above associated with low sample size). This protein, which was not up-regulated in Houwan corals exposed to this same temperature regime, has been suggested to function in vesicle docking-related exocytosis (including a role in the SNARE complex) and protein secretion [40]. Regarding the former process, lipid and membrane trafficking within the anthozoan-dinoflagellate endosymbiosis has been shown to be critical to the stability of these mutualisms [41,42], and Symbiodiniaceae synthesize, and later secrete, LBs into host cells [43]. Perhaps there is a link between the high photosynthetic output of these samples (Table 1) and concentration of this protein, both of which correlated with the higher growth rates observed (R^2^>0.25). For instance, higher photosynthetic rates could have led to elevated levels of Symbiodiniaceae LB synthesis and, consequently, LB trafficking from endosymbionts to hosts. Certainly, the idea of differential lipid trafficking within holobionts exposed to thermodynamic conditions (sensu [44]) is a worthy avenue for future research. In contrast to the relatively well developed understanding of the lipid dialogue between anthozoan hosts and dinoflagellates [45], less is known about the secondary role of sec1a, protein secretion, in anthozoan-dinoflagellate endosymbiosis, and others [46] have made a case for labeling and tracking protein flow within the holobiont; such an approach is also advocated here.

Although 2DGE was more likely to identify temperature-responsive proteins, site of origin effects were more evident upon multivariate assessment of the iTRAQ data. This is so even despite few individual proteins differing in concentration between corals from upwelling and non-upwelling reefs. Such upwelling reef vs. non-upwelling coral proteomic differences were maintained even after one week of exposure to a non-native temperature regime (in half the cases); in other words, the concentrations of numerous proteins showed “site fidelity.” Others have documented site fidelity in coral [47] and Symbiodiniaceae [48,49,50,51,52] gene expression patterns, which can persist even across large spatio-temporal scales [53]. Since corals herein were only maintained in laboratory aquarium culture for several weeks (including the acclimation period), it would be interesting to see if such site fidelity is eventually lost after longer periods of time in a common garden setting or if these site-associated proteome-scale differences are truly entrained in their biology. The observation that one Houwan sample from the variable treatment became more “Houbihu-like” may lend credence to the former notion, though this sample could simply represent an outlier demonstrating aberrant molecular physiology (sensu [54,55]). Although only *Cladocopium* spp. dinoflagellates were found in these colonies in situ, as well as in samples sacrificed after the termination of the experiment, it is also possible that within-lineage endosymbiont shuffling was responsible for driving some of the aberrant behavior associated with the Houbihu-like Houwan sample. Indeed, the role of endosymbiont community shifts on host and dinoflagellate proteome biology should be addressed in future studies.

It is worth further commenting on the differences between this study and another [31] that analyzed these same protein samples with 2DGE. It was hypothesized that, despite issues with linking spot intensity to protein concentration in 2DGE, at least several findings to have emerged from the older technology would have been confirmed using a next-generation proteomics approach, but aside from the LB and lipid trafficking proteins, the overlap was low (Appendix A). Part of this stems not strictly from the lack of sensitivity of 2DGE but actually of one major limitation of iTRAQ: low labeling efficiency. Herein, as well as in prior works with this approach [56], only 10–20% of proteins are typically labeled with the quantification tag. This means that 80–90% of the proteins, while sequenced by the MS, were not truly quantified. Perhaps, then, had better labeling efficiency been achieved, greater overlap between this study and the previous would have been documented. Despite such differences between the 2DGE and iTRAQ datasets, neither approach demonstrated a relationship between gene expression and protein concentrations. In the 2DGE analysis [31], we estimated that 2–10% of mRNAs encoded a protein that showed the same experimental trend, whereas herein that percentage was 0%, and the mean R^2^ between gene and protein concentration was effectively 0. This signifies that mRNA levels cannot be used to confidently predict protein concentrations or make cellular inferences for this reef coral or its endosymbiont communities. Nevertheless, mRNA biomarkers may still play a role in reef coral diagnostics provided their expression is tightly linked to later physiological changes that manifest in stressed or bleaching-prone corals (sensu [49,50,51]), regardless of whether or not the respective proteins follow suit.

The fact that, of the original 5000+ proteins sequenced by MS, only 30 passed all QC (of which only 1–2 of 4 RSA-screened putative DCPs were found to be differentially concentrated by more conservative post-hoc approaches), may be construed as a methodological failure, especially given the high costs of iTRAQ (nearly $200USD per sample, including nano-LC, MS/MS, and rudimentary data processing); indeed, an improvement in labeling efficiency must be achieved. Furthermore, despite the use of a normalizing sample to attempt to limit batch effects, a low percentage of proteins was sequenced in both batches. Given this finding, it is recommended that those looking to use iTRAQ in the future should analyze samples in pairs of four (e.g., four stable-temperature samples vs. four variable-temperature samples) within each batch, then carrying out statistical analyses for each batch as a repeated measure and comparing findings to those from other batches analyzed similarly. Although different proteins may nevertheless be sequenced across batches, this strategy will potentially enhance the number of within-batch, treatment-derived differences in protein concentrations, of which some putative DCPs will ideally be identified in multiple iTRAQ batches.

Despite the low number of proteins featured in the final suite of analyses, for a modeler looking to make predictions about field coral behavior, the need to incorporate only 2–4 proteins in a quantitative algorithm for determining how a coral will respond at the protein level to variable temperature exposure is actually a desirable outcome; why measure the concentrations of 20,000 proteins when the data from only a handful are needed to predict with 100% confidence how a coral will respond to a given environmental challenge (upwelling in this case)? Such significant site fidelity with respect to protein concentrations was documented herein that the proteomic data could be used to (1) define a protein signature for corals of each site and (2) predict how they would respond to variable temperature exposure. Regarding the latter feature, endosymbionts from Houbihu and Houwan corals maintaining high levels of sec1a and sec34, respectively, were associated with physiologically enhanced hosts at variable and stable temperature conditions, respectively, meaning that these proteins could play a role in coral thermotolerance, particularly with respect to upwelling events [57]. Furthermore, those looking to genetically modify host-endosymbiont assemblages [58] may benefit from further exploring the role of these proteins and more generally, holobiont lipid metabolism and trafficking, in the thermo-stability of the coral-dinoflagellate endosymbiosis [59].

## Figures and Tables

**Figure 1 microorganisms-08-01171-f001:**
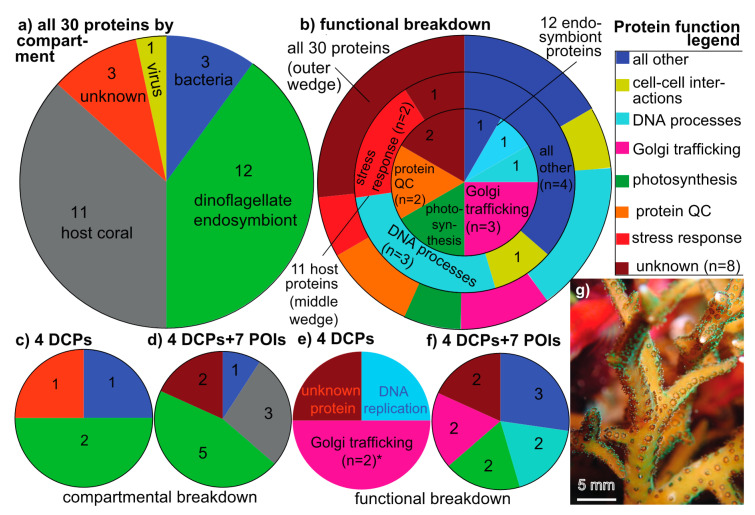
Compartmental and functional breakdown of the sequenced proteins, including those that were most differentially concentrated across experimental treatments (DCPs). Compartmental breakdown (**a**) of the 30 proteins that passed all quality control criteria: host (grey), Symbiodiniaceae dinoflagellates (green), bacterial (blue), viral (yellow), and unknown (red). These colors are used in remaining panels depicting compartmental differences (**c–d**), as well as in all following figures. Colors of the wedges in the DCP panels (**b**,**e**,**f**) instead correspond to the functional categories found in the “Protein function legend.” The lone functional category that was over-represented (Fisher’s exact test, *p*<0.01) in the most differentially concentrated proteome (**e**) versus the composite proteome (**b**) has been denoted by an asterisk (*). A macro image of a *Seriatopora hystrix* colony (**g**) from Komodo National Park (Indonesia). POI=protein of interest (see main text for definition.).

**Figure 2 microorganisms-08-01171-f002:**
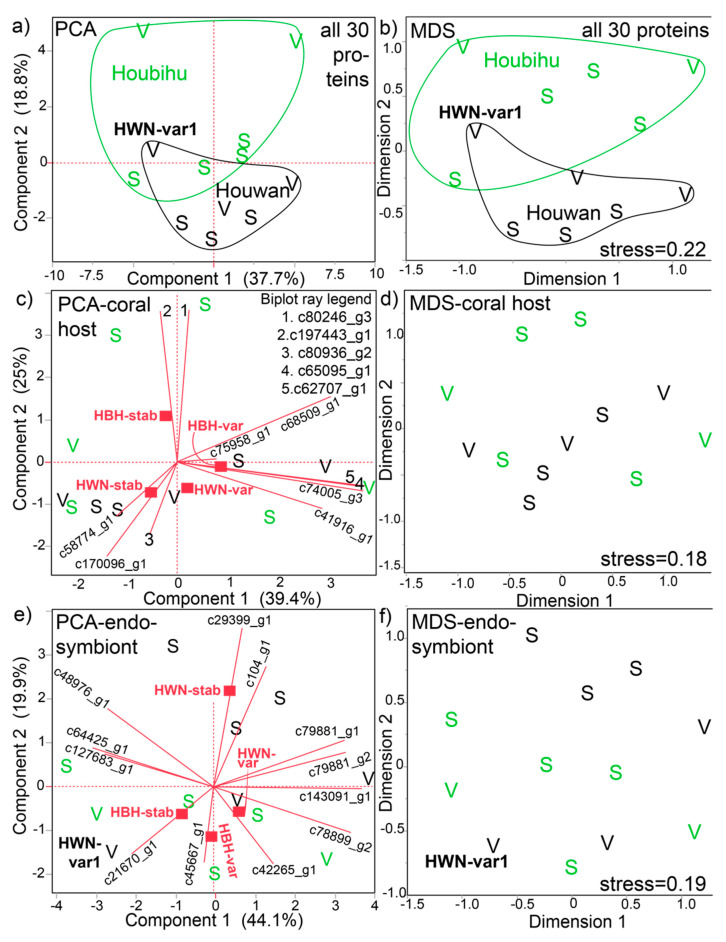
Multivariate analysis of the partial *Seriatopora hystrix*-Symbiodiniaceae proteome. Both principal components analysis (PCA) on correlations (**a**,**c**,**e**) and multi-dimensional scaling (MDS) with standardized data (**b**,**d**,**f**) were carried out with all 30 proteins (**a** and **b**, respectively), the 11 host coral proteins only (**c** and **d**, respectively), and the 12 endosymbiont proteins only (**e** and **f**, respectively). In panel **a**, samples from each site of origin, Houbihu (HBH; green icons and lines) and Houwan (HWN; black icons and lines), have been grouped by their k-means cluster; one HWN sample, HWN-var1, that fell within the HBH cluster has been labeled for emphasis in this panel, as well as certain others. The ellipses in panel **b** were, in contrast, drawn by eye and do not signify clustering. Certain biplot rays in panel **d** have been enumerated due to spatial constraints, with the legend found in the top-right corner of the panel. In the compartment-specific PCAs (**c**,**e**), the mean positions of the four site of origin x temperature treatment interaction groups have been shown as red squares. S=stable temperature samples. V=variable temperature samples.

**Figure 3 microorganisms-08-01171-f003:**
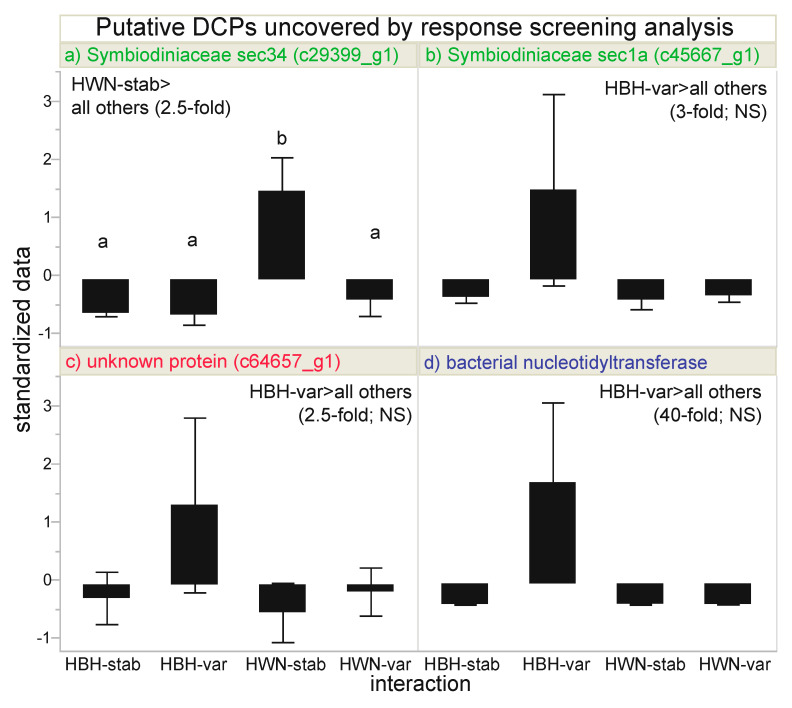
“Most differentially concentrated” proteins. Four proteins were identified by response screening analysis to be potentially differentially concentrated across the interaction of site (Houbihu [HBH] vs. Houwan [HWN]) and temperature regime (variable [var] vs. stable [stab]) at a false discovery rate-adjusted *p*-value of 0.01, and lowercase letters above standardized data in panel **a** represent significance (Tukey’s HSD post-hoc tests of box-cox-transformed data, *p*<0.05). Error bars represent standard deviation (*n*=2–4 per interaction group). Note that there were no pairwise differences for sec1a (**b**; box-cox-transformed data), the unknown protein c64657_g1 (**c**; log-transformed data), or the bacterial nucleotidyltransferase (**d**; non-parametric 2-way ANOVA). NS=not significant.

**Figure 4 microorganisms-08-01171-f004:**
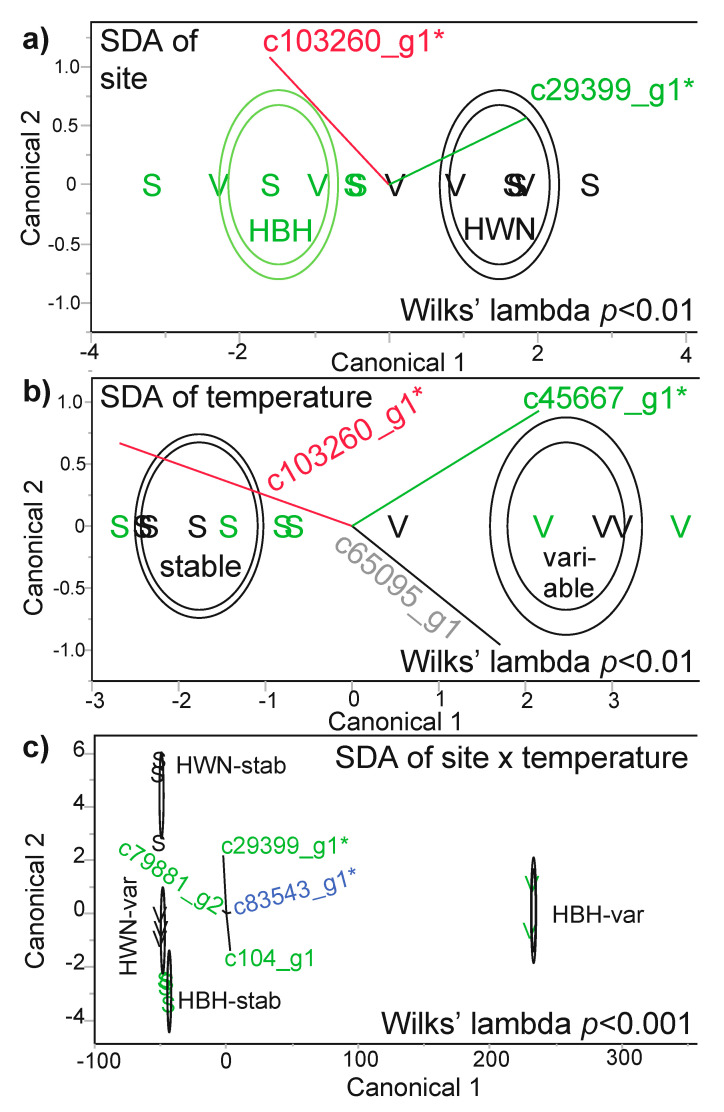
Stepwise discriminant analysis (SDA). The results of SDA using all 30 proteins as predictors for site of origin (**a**), temperature treatment (**b**), and their interaction (**c**). In all cases, the model built from the incorporated proteins (grey, green, blue, and red biplot rays for coral host, Symbiodiniaceae, bacterial, and proteins of unknown origin, respectively) correctly predicted the identity of all 12 samples (0% misclassification rate). Putative differentially concentrated proteins (DCPs) identified by response screening analysis (Figure 3) have been denoted by asterisks, and inner and outer contours represent 50 and 95% confidence, respectively. In certain cases, the stable (stab; S) and variable (var; V) temperature icons overlap, and Houbihu (HBH) and Houwan (HWN) icons are highlighted in green and black font, respectively. In panel **c**, some of the icons are masked by the contours, and the horizontal vectors (i.e., c79881_g2 and c83543_g1) are relatively compressed. Please also note that the significant MANOVA interaction effect in panel **c** should be interpreted cautiously given the low sample size of the HBH-var interaction group (*n*=2).

**Figure 5 microorganisms-08-01171-f005:**
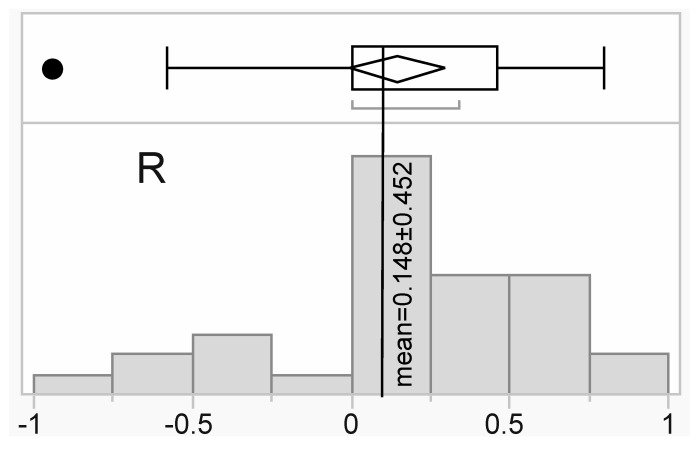
Histogram of gene expression vs. protein concentration correlation coefficient (R) values. The global mean has been shown as a vertical line, and the error terms represents standard deviation.

**Table 2 microorganisms-08-01171-t002:** Samples analyzed by transcriptomic (RNA-Seq [31]) and proteomic (iTRAQ; herein) approaches. Please note that (1) six samples were analyzed by both methodologies, and (2) all colonies were of the same genotype. NA=not applicable. temp.=temperature.

Sample Code	Temp.	Site of Origin	Interaction	Tank	RNA-Seq?	iTRAQ?	iTRAQ Batch-Label (Quantity)
HBH-var1	variable	Houbihu	HBH-var	HBH-V1	yes	yes	B-114 (18 µL)
HBH-var2	variable	Houbihu	HBH-var	HBH-V2	yes	no	NA
HBH-var4	variable	Houbihu	HBH-var	HBH-V2	yes	no	NA
HBH-var6	variable	Houbihu	HBH-var	HBH-V3	no	yes	A-114 (18 µL)
HBH-stab1	stable	Houbihu	HBH-stab	HBH-S1	no	yes	B-115 (22 µL)
HBH-stab3	stable	Houbihu	HBH-stab	HBH-S2	yes	yes	B-116 (18 µL)
HBH-stab4	stable	Houbihu	HBH-stab	HBH-S2	yes	no	NA
HBH-stab5	stable	Houbihu	HBH-stab	HBH-S3	yes	yes	A-115 (22 µL)
HBH-stab6	stable	Houbihu	HBH-stab	HBH-S3	no	yes	A-116 (18 µL)
HWN-var1	variable	Houwan	HWN-var	HWN-V1	yes	yes	A-117 (17 µL)
HWN-var2	variable	Houwan	HWN-var	HWN-V1	yes	no	NA
HWN-var3	variable	Houwan	HWN-var	HWN-V2	yes	yes	B-119 (20 µL)
HWN-var5	variable	Houwan	HWN-var	HWN-V3	no	yes	B-117 (17 µL)
HWN-stab1	stable	Houwan	HWN-stab	HWN-S1	yes	yes	B-118 (18 µL)
HWN-stab3	stable	Houwan	HWN-stab	HWN-S2	yes	no	NA
HWN-stab4	stable	Houwan	HWN-stab	HWN-S2	no	yes	A-118 (18 µL)
HWN-stab5	stable	Houwan	HWN-stab	HWN-S3	yes	no	NA
HWN-stab6	stable	Houwan	HWN-stab	HWN-S3	no	yes	A-119 (20 µL)

**Table 3 microorganisms-08-01171-t003:** Results of stepwise regression (SRA) for identifying “proteins of interest” (POIs). Putative differentially concentrated proteins identified by response screening analysis have been denoted by asterisks (*); the other five proteins instead represent POIs (with those identified by stepwise discriminant analysis underlined). Please note that no bacterial or viral proteins featured in any of the SRA models. Symbiodiniaceae (Sym) proteins: c104_g1=“in between ring fingers” protein (protein-protein interactions) and c79881_g1=peridinin chlorophyll a-binding protein (photosynthesis). Host coral proteins: c197443_g1=Pao retrotransposon peptidase (DNA modification) and c75958_g1=zinc finger CCCH domain-containing protein 3-like (mRNA export). All other proteins could either not be identified (c103260_g1) or are described in Figure 3. BIC=Bayesian information criterion.

Experimental Factor	#Proteins	BIC	Host Coral Proteins	Sym Proteins	Unknown Proteins
Site of origin	2	7.45		c29399_g1 *	
				c104_g1	
Temperature (temp.)	2	16.3		c45667_g1 *	c103260_g1
Site of origin x temp.	4	17.4	c197443_g1	c79881_g1	c103260_g1
			c75958_g1		

**Table 4 microorganisms-08-01171-t004:** Summary of most differentially concentrated proteins (DCPs) and “proteins of interest” (POIs). None of these proteins were identified in prior 2-dimensional gel electrophoresis-based analyses. Please note that the two peridinin-chlorophyll A-binding protein (PCP) isoforms differ slightly in sequence (see online supplemental data file for exact peptide sequences.). For a graphical depiction of the compartmental and functional breakdown, please see Figure 1. HBH=Houbihu (upwelling site). HWN=Houwan (non-upwelling site). ND=no difference. QC=quality control (processing, modification, and nascent folding). RSA=response screening analysis. SDA=stepwise discriminant analysis. SRA=stepwise regression analysis. Stab=stable temperature regime. Sym=Symbiodiniaceae. Var=variable temperature regime. * Marginally significant difference (0.01<*p*<0.05 [non-false discovery rate-adjusted]).

Accession	DCP/ POI	Compart-ment	Protein (Figure)	Protein Function	RSA Trend	SDA Models	SRA Models
c29399_g1	DCP	Sym	sec34 (3a)	Golgi trafficking	HWN-stab>all others	site, interaction	site
c45667_g1	DCP	Sym	sec1a (3b)	Golgi trafficking	HBH-var>all others ^a^	temperature	temperature
c64657_g1	DCP	unknown	unknown (3c)	unknown	HBH-var>all others ^a^		
c83543_g1	DCP	bacteria	nucleotidyltransferase (3d)	DNA replication	HBH-var>all others ^a^	interaction	
c103260_g1	POI	unknown	unknown	unknown	ND	temperature	temperature, interaction
c79881_g1	POI	Sym	PCP (g1)	photosynthesis	HWN>HBH *		interaction
c104_g1	POI	Sym	ring finger	protein QC	ND	interaction	site
c197443_g1	POI	host	Pao retrotransposon peptidase	DNA modification	ND		interaction
c75958_g1	POI	host	zinc finger CCCH domain- containing protein 3-like	mRNA processing	ND		interaction
c65095_g1	POI	host	calmodulin	calcium regulation	ND	temperature	
c79881_g2	POI	Sym	PCP (g2)	photosynthesis	HWN>HBH *	interaction	

^a^ Significant RSA interaction effect in overall model, but no post-hoc differences among individual means upon box-cox-, log-, or rank-transforming the data.

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
