# Peer review of "Proteomic Signatures of Corals from Thermodynamic Reefs"

_microorganisms, 2020, doi:10.3390/microorganisms8081171_

Round 1

Reviewer 1 Report

In this paper,  the author is trying to identify the proteome of one species of coral in  Taiwan, that experience very different temperature regimes at two different sites. He used several methods to find proteins with  concentrations that differ significantly among sites. Author found 4 such proteins, (2 from the coral symbiont, one unknown, one bacterial).

I find the general question interesting and worth pursuing. However, this paper is hard to read. It has so many abbreviations, and long and convoluted sentences, it  needs major style reassessment.

The author name is displayed  incorrectly on the front page of the article  (Mayfield and Anderson B.).

It does not help that the author continuously refers to other papers when it comes to the details of the methods. The paper needs to be self contained, so please make sure it is so. Readers need to be able to read the paper and assess all relevant parts without reading several additional papers (see for example paragraph 2.1).

I am confused by table 1. It is supposed to show the “physiological data” in hand. What I see is a * (or no *)  for temperature, site, inc. ( what is inc.?) what does it mean? I understand what the abbreviations mean, but what does HWN>HBN mean? This table needs an explanation (or a better one of what is presented).

Line 95 -97: I am not sure what this sentence means. Please rephrase. Similarly following sentences down to the end of the paragraphs are obscure.

Line 102. This sentence is 8(!) lines long. Please, break it down in several sentences. It is very hard to follow the author’s thoughts.

Please end introduction with clear aims. “This paper aims to …..”.

Paragraph 2.1. corals where acclimated  in indoor aquaria. Please specify conditions. Specify when the corals where collected. Was it an upwelling season? What were the exact environmental conditions when collected? Which species ? How many colonies? How many different sites?

I am also confused by Table 1 that is reported in the intro, but also as results. What is of the two? Are these data collected in the paper or from previous studies?

Line 142. “the 12 proteins”. Which 12 proteins is the author referring to?

Overall, the results are scarce, with only 4 proteins found to be differentially concentrated, none of them from the coral, and one unknown.  I doubt these methods are effective in answering the questions at hand.

I believe that, given the limited results, and the overall shortfall of the paper, and the limited contribution of this work,   this paper cannot be accepted in the journal of Microorganisms.

Author Response

Please see attached document (it includes some figures). 

Reviewer 2 Report

This work is an important addition to identifying the mechanisms of coral bleaching and why some coral bleach with increasing temperature. The difficulties of these experiments, growing coral and the multi-omic data needed to identify the mechanisms makes this work difficult. Pharma companies spend $100s of millions doing similar multi-omic data fusion studies to identify mechanisms of disease. While this experiment is under powered with sample numbers, expense and funding are the clear constraints on a thorough and well worked up study. Expanding the transcriptomic survey sample size and sequencing depth may help, even if expression is almost orthogonal to protein abundance. Fusing data from multiple experimental platforms is certainly a very good approach to pathway discovery in this application, and the author is commended for such thorough work with the limited means available to him.

Author Response

Thank you for your positive comments, and I am glad you appreciate the work that went in with a shoe-string budget. I actually used charitable donations to fund the proteomics! You make a good point that the transcriptome was sequenced with Illumina's Tru-Seq so long ago (2010) that the transcriptome now pales in comparison in size and coverage to what people are getting using 2020 chemistries and instruments. As it turns out, despite essentially "negging" transcriptomics in the manuscript, I myself still have transcriptome projects in the works because I think they may still have value. Thank you again for reading and endorsing this work. 

Reviewer 3 Report

This manuscript examines the proteomic response of Seriatopora hystrix to stable or variable temperature regimes, in the context of the coral’s native reef environments in upwelling and non-upwelling areas in Taiwan. The author uses iTRAQ to characterise proteins isolated from Houbihu (upwelling) and Houwan (non-upwelling) corals exposed to either stable or variable (6 °C range over 6-h cycle) temperatures. The experiment led to 30 proteins that passed QC across all treatments, and showed that corals exposed to their native temperature environments had the highest growth rates and were most different in their proteomes. An important differentially concentrated protein is the algal endosymbiont lipid trafficking protein, sec1a, which was up-regulated only in upwelling corals exposed to variable temperatures and could underlie acclimation to thermal extremes.

Studies like these are important because proteomic analyses are still quite a way behind gene expression studies in unravelling the cellular mechanisms behind corals’ responses to temperature change. Given that proteomic patterns tend to be distinct from mRNA changes, the present characterisation will be a very meaningful contribution and I hope to see it in print to add to the growing body of work in this field.

A major concern I have concerns the replication used in the experiment. Analysis of 2–4 colonies per origin site and treatment is obviously inadequate, and the author acknowledges this. However, if the inferences are made in more qualitative terms, the study still offers valuable insights. In particular, I urge the author to relook at the comparisons among sites and treatments. I don’t think parametric tests like Tukey test are acceptable, and the presentation of quantiles (e.g. Figure 3) is clearly inappropriate for 2–4 points of each protein response.

Additionally, it’s not clear how the replicates in each treatment are aligned across treatments in terms of the genotypes used. Table 2 suggests that nubbins from the same colonies are tested across treatment. If so, then they need to be paired; they are clearly not independent. But it also appears they are randomly chosen in each origin site and treatment, which means that only some are clones tested across treatments. These issues need clarification and possibly a simplification in how some of the analyses are approached.

One major point the author makes is that there is no underlying relationship between gene expression and protein concentration that is predictable, so that transcript levels and transcriptomes cannot be used to understand protein concentrations. I agree, but as cellular responses are ultimately initiated at the gene level—which does have substantial influence on downstream cellular processes, albeit not with the simple, independent, linear functions tested here—I think the judgement above has been passed too quickly. Some genes are more responsive than others (see e.g. Levy et al. 2011; http://doi.org/10.1126/science.1196419) and for different processes and timescales (e.g.
Cziesielski et al. 2018; https://doi.org/10.1098/rspb.2017.2654). As explained by Cleves et al. (2020; https://doi.org/10.1016/j.tig.2019.11.001) and others, the solution to understanding coral functions and survival is a multi-omics one. The proteome is important, but is a part of the cellular machinery which needs many linked tools to characterise, including genomics, transcriptomics, proteomics, and epigenomics.

Following are some specific points:

Lines 16–22: Please break this sentence into two, one for the approach and one the result.

Line 126: What do you mean by ‘The genotypes of both Houbihu and Houwan corals were identical based on analysis of microsatellites’? That they are clones within/among sites? Or they are genetically undifferentiated within/between sites?

Lines 118, 130 and 139: Please confirm that these 2–4 nubbins per site-treatment may or may not be the same genotype across treatments? i.e. some genotypes are tested across treatment while others are not? If so, I assume the analyses are not paired (or linked by genotypes)? Clarity is needed on the replication (if any within a genotype) and genotypes, so that it’s clear how the variances are partitioned across treatment, genotypes and nubbins.

Line 158: This should be S. hystrix holobiont transcriptome.

Line 201: The number of samples is rather small for each site-treatment (n=~3, not 12, and unbalanced), so there is a bit of concern about the power to detect differences using PERMANOVA. Please perform some form of power analysis to check (e.g. Kelly et al. 2015; http://doi.org/10.1093/bioinformatics/btv183).

Line 217: Some clarification is needed on the definition of POIs—these are ‘proteins whose concentrations differed significantly among site of origin (df=1), temperature treatment (df=1), and/or their interaction’, but they may not be predictive of coral behaviour in the RSA?

Section 2.6: Check and state the assumptions of SDA and SRA with respect to parametric and sample size requirements.

Line 234: This is a really low proportion of the proteins detected (and passing QC) across iTRAQ batches, and is also absolutely low with respect to typical studies. What would account for this large variation between batch A and B? The utility of such low numbers of proteins is explained for future analyses in lines 475–488, but the ‘why’ question has not been addressed.

Lines 286–293: When considering the biplots particularly with respect to the PCA, component 1 typically explains up to twice that of component 2. Therefore, the visual characterisation of the clustering needs to take into account the pattern that within site-treatment variation is often along component 1, so there is limited segregation in a multivariate sense. In the context of the NMDS, if all the variation is collapsed into 1 dimension, it may look rather indistinguishable between sites and between treatments.

Figure 3: The general point is explain above, but specifically for the figure, there is no reason to draw the (trend) lines between bars (site-treatment). The use of quantiles is also misleading since there are only 2–4 points. I suggest presenting just the points. The use of Tukey’s test is also suspect since normality is highly unlikely.

Lines 451: Please explain if there is a role for community and/or population switching/shuffling of Symbiodiniaceae that may influence gene expression, primarily in the endosymbionts but also in the host?

Author Response

Please see attached document (which is actually the entire rebuttal letter, not just for reviewer #3). 

Round 2

Reviewer 3 Report

I appreciate the author's frank discussion and addressing most if not all of the concerns. I believe it will be a good contribution to the field.

I would like to reiterate that Figure 3 should do away with the lines between site-treatments simply because they are distracting and also misleading in suggesting a trend.

There is also a difference in saying that 'The genotypes of both Houbihu and Houwan corals were later found to be identical based on analysis of microsatellites' and that they are clones. As the author points out, higher-resolution methods could result in distinction of genotypes, so it should be clear in the text that the analyses effectively treats the colonies as clones.

Author Response

Reviewer #3’s comment #1: I would like to reiterate that Figure 3 should do away with the lines between site-treatments simply because they are distracting and also misleading in suggesting a trend.

Author response to reviewer #3’s comment #1: Although they are meant to show interactions of site and temperature, I can see where some people associate lines with temporal trends. Therefore, I have changed them to bar graphs. I had wanted to avoid using bar graphs since the data were standardized, meaning some values are negative (which might appear confusing to a reader not prepared to see what may be perceived as negative protein concentrations). However, it is the relative differences among treatments that is key, in any event, and bar graphs show such differences without implying trends.

Reviewer #3’2 comment #2: There is also a difference in saying that 'The genotypes of both Houbihu and Houwan corals were later found to be identical based on analysis of microsatellites' and that they are clones. As the author points out, higher-resolution methods could result in distinction of genotypes, so it should be clear in the text that the analyses effectively treats the colonies as clones.

Author response to reviewer #3’2 comment #2: Yes. In all honesty, I bet they ARE different genotypes, but since the data do not show this, I have taken the most parsimonious conclusion of assuming the microsat data are legitimate. Anyway, that’s why that’s essentially an obsolete technology! I have expanded on the following sentence because I think this is a key point of this manuscript (that it’s essentially phenotypic plasticity until proven otherwise). New lines 154-159:

“Although future genetic analysis featuring higher resolution approaches, such as next generation nucleic acid sequencing, may ultimately uncover genetic differentiation among the Taiwan Strait (Houwan) and South China Sea (Houbihu) corals, clonality has been assumed herein; this signifies that all variation in physiology and proteome biology documented is presumably from environmentally-driven phenotypic plasticity (i.e., acclimatization) and not a result of adaptation (assuming microbial assemblages to be similar among them).”